# Hierarchical Prompting Assists Large Language Model on Web Navigation

**Abishek Sridhar**[*]   **Chi-Fan Lo**[*]   **Frank F. Xu**   **Hao Zhu**   **Shuyan Zhou**[†]

School of Computer Science, Carnegie Mellon University

{abisheks,chifanl,fangzhex,zhuhao,shuyanzh}@cs.cmu.edu

## Abstract

Large language models (LLMs) struggle on processing complicated observations in interactive decision making tasks. To alleviate this issue, we propose a simple hierarchical prompting approach. Diverging from previous prompting approaches that always put the *full* observation (*e.g.,* a web page) to the prompt, we propose to first construct an action-aware observation which is more *condensed* and *relevant* with a dedicated SUMMARIZER prompt. The ACTOR prompt then predicts the next action based on the summarized observation. While our method has broad applicability, we particularly demonstrate its efficacy in the complex domain of web navigation where a full observation often contains redundant and irrelevant information. Our approach outperforms the previous state-of-the-art prompting mechanics by 6.2% on task success rate, demonstrating its potential on interactive decision making tasks with long observation traces. [1]

## 1 Introduction

In our daily lives, we often encounter tasks such as household duties (Shridhar et al., 2020) and web navigation (Shi et al., 2017; Zhou et al., 2023) that necessitate interactive and sequential decision-making. These tasks require us to take actions (*e.g.,* entering a search query) based on both the state of the environment and the specific objectives (*e.g., buy a shirt*). There has been growing interest in automating these decision-making tasks using natural language commands (Yang et al., 2023).

Large language models demonstrate their potential on performing interactive decision making tasks, owing to their capacity for encoding vast amounts of real-world knowledge. One common approach for utilizing LLMs in these tasks is

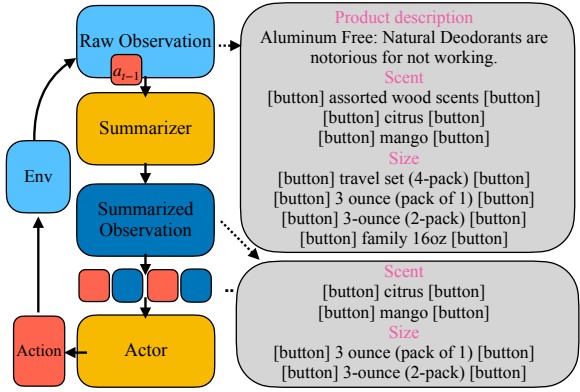

Figure 1: ASH Prompting. In each round, the SUMMARIZER will produce a condensed observation given raw observation and the ACTOR's previous action. The ACTOR will produce the next action based on the more concise summarized observation and the interaction history. The right side shows the raw observation and the summarized observation for the instruction "*Find me a small bottle of fruit deodorant*".

through few-shot in-context examples (Ahn et al., 2022). Previous research has focused on various techniques for shaping the *action space*, such as structured representations of actions through programs (Zhou et al., 2021; Liang et al., 2022; Singh et al., 2022), and promoting synergy between reasoning and acting (Yao et al., 2022b). However, little attention has been paid to effectively encoding the *states*, which is particularly crucial when the interacting environment is diverse and lengthy.

This work aims at optimizing the state observation with prompting method. We propose ASH (Actor-Summarizer-Hierarchical) prompting as in Figure 1. Our approach has two key components: (1) A SUMMARIZER that takes a raw observation from the environment and produce a new representation that is more *meaningful* to the goal. It achieves this by learning from a dedicated prompt that demonstrates the expected summarization in different scenarios. In Figure 1, the instruction requests a small bottle of deodorant with a fruit

---

[*]Equal contribution

[†]Corresponding author

[1]Code is available at https://github.com/robert1003/ash-prompting

scent. The current product includes suitable configurations that meet this requirement. However, there are unnecessary details like product descriptions and distracting options such as large-volume bottles. The summarizer filters out these irrelevant contents, retaining only the relevant information for further processing. (2) An ACTOR that takes the trajectory history (*i.e.,* observations and actions) and produce the next action. The hierarchical modularized design of ASH effectively reduce the heavy reasoning burden by avoiding dumping *all* information to a single prompt. The SUMMARIZER of ASH could effectively remove irrelevant information, and provide a more concise observation representation. This representation aligns with the demonstration in the context for the actor to make more precise predictions.

We apply ASH on web navigation with a special focus on purchasing products on a real-world scale E-commerce simulator Webshop (Yao et al., 2022a), since it is the only relevant existing dataset with a complex observation space to the best of our knowledge. Our ASH significantly outperforms the previous state-of-the-art approach (Yao et al., 2022b) by 6.8% on task success rate (29% relative gain). These results show the promise of ASH in addressing the challenges of comprehensive history and complicated states in real-world interactive decision making tasks.

## 2 The Web Navigation Task

In this section, we formally define the task of web navigation. Given a web environment $\mathcal{E}$, initial state $s_0$ with observation $o_0$ (typically a web page), the goal of an agent is to reach a specific environment state $s^*$ defined by natural language $u$ by performing a sequence of actions. At time $t$, the agent needs to decide the action $a_t$ based on interaction history up to $t$ ($H_t = \{o_0, a_1, o_1, \ldots, a_{t-1}, o_t\}$). Once a stop action is predicted, the score of the task is measured by comparing the final state $\hat{s}$ with $s^*$ with a similarity function $f(\hat{s}, s^*)$ which ranges from zero to one.

## 3 ASH Prompting

As summarized in Figure 1, ASH comprises of a stateless SUMMARIZER that takes an action $a_{t-1}$ and observation $o_t$ and then outputs a condensed observation $o'_t$ that is semantically comprehensible by the ACTOR. Then, the stateful ACTOR takes updated history $H'_t = \{o'_1, a_1, o'_2, a_2, \ldots, a_{t-1}, o'_t\}$

```
Action: click[B078GWRC1J]

Instruction: Find me a 3 ounce bottle of
of bright citrus deodorant

Observation:
scent
[button] assorted scents [button]
[button] bright citrus [button]
size
[button] travel set (4-pack) [button]
[button] 3 ounce (pack of 1) [button]
[button] 3-ounce (2-pack) [button]
...

Information:
Let us think step-by-step.
First, I need to let the actor
know if this is the correct product.
Then, I need to filter out options
that does not match to make it easier
for the actor to choose from

scent
[button] bright citrus [button]
size
[button] bright citrus [button]
[button] 3 ounce (pack of 1) [button]
[button] 3-ounce (2-pack) [button]
```

Figure 2: An example in the SUMMARIZER prompt. The history information is in red; the raw observation is in blue ("..." indicates the omission of the full content); CoT instruction for the SUMMARIZER is in green. The SUMMARIZER's output is in black.

and predicts $a_t$. Formally, ASH decomposes the probability of generating $a_t$ to two components:

$$P\left(a_t | H_{t-2}, a_{t-1}, o_t\right) =$$
$$\underbrace{P\left(o'_t | a_{t-1}, o_t\right)}_{\text{(stateless) SUMMARIZER}} \cdot \underbrace{P\left(a_t | H'_{t-2}, a_{t-1}, o'_t\right)}_{\text{(stateful) ACTOR}}$$

### 3.1 SUMMARIZER

The goal of the SUMMARIZER is to generate a more concise observation that only encodes the *relevant* information. It identifies and removes irrelevant information is guided by the provided in-context examples. An example of the SUMMARIZER prompt is in Figure 2 and the full prompt is listed in Appendix C.1 and C.2.

There are several key information provided in the prompt for SUMMARIZER: (1) the previous action and the instruction (in red) that gives SUMMARIZER hints on *how* and *why* does the agent navigate to the current state. (2) raw environment observation (in blue) at current step that needed to be summarized. In web navigation task, this corresponds to the contents of the current web page. (3) a customized step-by-step instruction (Wei et al.,

```
Instruction: Find me a 3 ounce bottle of
of bright citrus deodorant

History:
Action 1:
search[3 ounce bright citrus
deodorant sensitive skin]

Observation 1:
Search results for keyword "3 ounce..."
[B078GWRC1J]
Bright Citrus Deodorant...
$10.99

...

Action 4:
click[3 ounce (pack of 1)]

Observation 4:
You have clicked 3 ounce (pack of 1).

Action 5:
click[Buy Now]
```

Figure 3: An example in the ACTOR prompt. The NL instruction is in red; the history is in blue. The ACTOR's output is in black.

2022) on *how* to summarize the current state (in green). Importantly, we design specialized examples tailored to different scenarios to encourage versatile reasoning. For example, as in Figure 2, the instruction outlines how to process a product page. Specifically, the SUMMARIZER is tasked to verify the desirability of the product, followed by filtering out any irrelevant options. More instructions on improving search query, compressing search results and others can be found in Appendix C.1. In this way, the summarized states can be more customized towards the current scenario.

## 3.2 ACTOR

We follow existing works to design the ACTOR prompt. Each example is consist of the NL instruction, the interaction history up to the current time step $t$. The history is presented as an alternating sequence of actions and *summarized* observations. The goal of the ACTOR is to produce the desired action for the current step. An example of ACTOR is listed in Figure 3 and the full prompt is listed in Appendix C.3 and C.4.

## 4 Experiment

**Dataset** We test ASH on Webshop (Yao et al., 2022a), a real-world scale E-commerce benchmark where the agent interacts with an Amazon-style website to purchase products described by

some NL descriptions (*e.g., "Find me black e-reader with price lower than 270 dollars"*). In this benchmark, the agent can perform three actions: (1) search[...] with a text-based query (2) click[...] any interactable web element (3) think[...] the current situation and make further movements. The think action, which does not change the state, is adopted from REACT (Yao et al., 2022b). Webshop is challenging for intelligent agents due to its large-scale and diverse product catalog, consisting of $1,181,436$ unique products, each with its own distinct features, resulting in a highly complex and varied state space.[2]

**Task Score Calculation** We follow (Yao et al., 2022a) to calculate the task scores. If a trajectory ends with buying a product, it will be assigned a score between zero and one, denoting the percentage of required attributes satisfied by the product. A score of 1.0 indicates a successful trajectory, while a 0.0 marks a failure. However, as a special case, if a trajectory either produces $k$ consecutive invalid actions or exceeds the max length, it will be terminated and assigned a score of 0.0.

**Model and Hyperparameters** We experiment with CODE-DAVINCI-002[3] (Chen et al., 2021) and GPT-3.5-TURBO[4] (OpenAI, 2022). We use greedy decoding with a max token of 500. We terminate a trajectory after 20 steps (the average number of steps in Webshop is 6.5), or if the agent issues an invalid action (*e.g.,* click a button that does not exist) for five consecutive steps.

## 5 Results

The main result is given in Table 1. ASH consistently outperforms the previous state-of-the-art prompting mechanism REACT with different base models. More specifically, with CODE-DAVINCI-002, ASH brings a 6.8% gain on success rate over REACT (29% relative gain) and improve the average score by over 3.3%. ASH demonstrates a more significant improvement when used with GPT-3.5-TURBO. A primary reason for this difference

---

[2]We utilized the text-rich settings of the Webshop on the entire product space and sampled a consistent 500 trajectories for our experiments. Unlike (Yao et al., 2022b), we kept the webpage complexities and did not restrict search results to the top three. The trajectory generation code is in Appendix F.

[3]We found that the free CODE-DAVINCI-002 yields stronger performance than the paid TEXT-DAVINCI-002 hence we used the former for all our experiments. The comparison is shown in Appendix A.

[4]We used GPT-3.5-TURBO-0613.

is the context length limitations of $4k$ in GPT-3.5-TURBO (CODE-DAVINCI-002 offers a longer context window of $8k$). ASH summarizes observations and effectively encodes history within this limited context budget. In contrast, REACT often surpasses the context length, leading to failures. This trend is unchanged when using a GPT-3.5-TURBO actor with no in-context examples, proving the method' insensitivity to number of in-context examples. These results demonstrate the effectiveness of ASH on performing web navigation tasks. We illustrate REACT and ASH on a sample trajectory in Appendix B and analyze the behavior of SUMMARIZER qualitatively in Appendix E.

| Method | Avg Score | Success (%) |
|---|---|---|
| REACT$_{\text{CODE-DAVINCI-002}}$ | 53.4 | 23.4 |
| ASH$_{\text{CODE-DAVINCI-002}}$ | **56.7** | **30.2** |
| REACT$_{\text{GPT-3.5-TURBO}}$ | 27.3 | 3.0 |
| ASH$_{\text{GPT-3.5-TURBO}}$ | **44.5** | **12.6** |

Table 1: Overall performance of REACT and our proposed method (ASH) with CODE-DAVINCI-002 and GPT-3.5-TURBO.

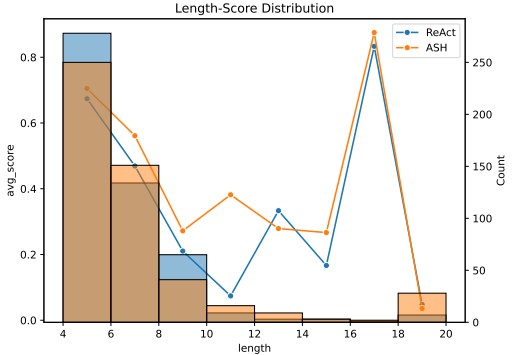

Figure 4: The average scores of trajectories grouped by the trajectory length, with the line plot representing score and the bar plot representing counts.

**Performance *vs.* trajectory length** Long-horizon tasks are challenging due to the difficulty of keeping track of long histories. To analyze this problem, we plot the average score with respect to the length of the trajectories in Figure 4. We observe that ASH consistently yields a higher average score than REACT across different trajectory lengths. The performance gap is particularly large on examples with more than 11 steps. While REACT only achieves an average score of 7.4, ASH increases the score to 38.2. This finding highlights the effectiveness of our SUMMARIZER module in

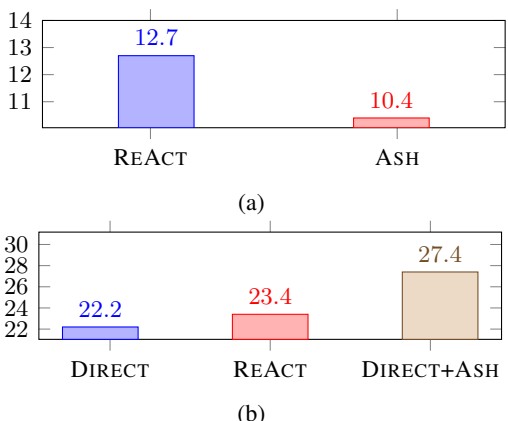

Figure 5: Top: percentage of failures due to repeating invalid actions. Bottom: success rate (%) on different prompting mechanisms.

generating more relevant and concise observations, which consequently enhances the ACTOR to make correct predictions in more complex tasks.

**ASH promotes valid action grounding** LLMs suffer from hallucinations even when evidence is provided within the context (Liu et al., 2023). In interactive decision making tasks, these hallucinations manifest as the generation of invalid actions that cannot be executed in the current state, such as attempting to click a non-existent button. As shown in Figure 5a, 12.7% of failures in REACT occur due to the repetition of invalid actions, resulting in termination because of the maximum step limit. In contrast, ASH exhibits a relative reduction of 18% in this failure mode, effectively decreasing the failure rate to 10.4%. Our hypothesis is that the presence of noisy and irrelevant context triggers hallucinations in LLMs, while providing a more concise and relevant context mitigates their impact.

**ASH mitigates the need of explicit reasoning** Previous studies such as REACT and CoT suggests that explicitly performing reasoning would assist more accurate decision making. However, we argue that the requirement of such verbose reasoning partially stems from LLM's inability to process complicated observations. To demonstrate this, we apply ASH to a simpler prompting mechanism that directly predicts the next action given the interaction history *without* the extra reasoning action (DIRECT). As in Figure 5b, we find that ASH would allow the naive prompting mechanism to outperform the more sophisticated REACT approach by a significant margin.

## 6 Related Work

**Prompting-based Decision Making**  Singh et al. (2022) and Liang et al. (2022) study the structured representation of actions. Instead of representing actions as a linear sequence, they turn actions to components in programs to encourage hierarchical structures (Zhou et al., 2021). Yao et al. (2022b) improve the action prediction by introducing an additional `think` step which verbally reasons about what needs to be done next. However, most works use the vanilla raw observations from the environment (Ahn et al., 2022; Huang et al., 2022). RE-ACT Yao et al. (2022b) is the work that is closest to us. However, ASH differs from REACT mainly in two ways: (1) ASH breakdown the observation processing into two steps: first to condense observations and then to act, while REACT do the first step internally and only act. (2) ASH separate the development of ACTOR and SUMMARIZER prompts, allowing for greater enhancement, whereas REACT solely depends on the ACTOR prompt's efficacy.

**Multi-stage Prompting**  Multi-stage prompting is explored in other tasks. (Liu et al., 2022) use multi-stage prompting model for knowledgeable dialogue generation. They show prompting again on the previously generated knowledge can help the model to generalize to context outside of the knowledge base. Similarly, by applying different prompts during multiple stages, (Tan et al., 2021) show that this simple approach can effectively adapt pre-trained LMs to downstream tasks like translation.

## 7 Conclusion

We introduce ASH, a hierarchical method tailored for sequential decision-making in web navigation using LLMs. Tested on the Webshop benchmark, ASH surpassed the strong prompting method RE-ACT in average score and success rate. Notably, ASH excels in complex tasks with lengthy trajectories. It reduces LLMs' hallucinations and mitigates the need of explicit reasoning.

## Acknowledgement

We thank the anonymous reviewers for valuable feedback. This material is partly based on research sponsored in part by the Air Force Research Laboratory under agreement number FA8750-19-2-0200. The U.S. Government is authorized to reproduce and distribute reprints for Governmental purposes notwithstanding any copyright notation thereon. The views and conclusions contained herein are those of the authors and should not be interpreted as necessarily representing the official policies or endorsements, either expressed or implied, of the Air Force Research Laboratory or the U.S. Government. This project was also partially supported by a gift from AWS AI.

## Limitations

The primary aim of this paper is to introduce a new approach of prompting. We did not employ detailed manual or automatic prompt engineering for the ACTOR and SUMMARIZER prompts. Another limitation is that we did not test our approach on more realistic benchmarks on web navigation, due to a lack of such benchmarks at the time we finished this work.

## Ethics Statement

Our work is on a novel prompting paradigm that could be applicable on various sequential planning tasks. To the best of our knowledge, this paradigm does not discriminate against or cause damage to any human. However, it is worth noting that automating web navigation with models poses potential risks such as unintentionally access or scrape sensitive or private information, leading to potential breaches of privacy.

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

# A  Performance on other LLMs

We initially tested on a smaller subset of 50 samples (out of 500) with contemporary models TEXT-DAVINCI-002 and CODE-DAVINCI-002. As evident in table 2, we observed that in both methods (REACT and ASH) CODE-DAVINCI-002 performs better than TEXT-DAVINCI-002, and a similar trend of ASH dominating REACT was evident in both models. Therefore, we proceeded to use CODE-DAVINCI-002 in our results and analysis.

Due to the unavailability of PaLM model (Chowdhery et al., 2022) for public access and the mildly obscure implementation details of ReAct on Webshop from the paper (Yao et al., 2022b), we re-implement our version of ReAct on CODE-DAVINCI-002 and GPT-3.5-TURBO, and design prompts for ASH in an analogous fashion to run our experiments.

| Method | Avg Score | Success |
|---|---|---|
| REACT + TEXT-DAVINCI-002 | 50.2 | 15.3 |
| ASH + TEXT-DAVINCI-002 | **60.9** | **23.1** |
| REACT + CODE-DAVINCI-002 | 61.3 | 23.1 |
| ASH + CODE-DAVINCI-002 | **62.6** | **28.8** |

Table 2: Overall performance (%) of REACT and our proposed method (ASH) with TEXT-DAVINCI-002 on 50 random sampled trajectories.

# B  REACT vs ASH Prompting Trajectory

We see an illustration of where REACT and ASH method differ in approaching an instruction. While REACT hurriedly picks a sub-optimal product without worrying about one desired property being off, ASH notices that the correct color is not in the product initially picked from the match information between instruction and description by the summarizer and proceeds to reformulate the query and buy the correct product. The actor also finds it easy to identify the optimal product based on the retention of fewer options by the summarizer, as we intuitively imagined. The example also demonstrates the effectiveness of ASH in reducing the hesitation of the actor to take longer trajectories.

## B.1  REACT

```
WebShop
Instruction:
Find me white item window coverings for living room, dining room with color: color13
    , and size: 84"w x 84"l, and price lower than 150.00 dollars
[Search]

Action 1: search[white item window coverings for living room dining room]
Observation 1:
Instruction:
Find me white item window coverings for living room, dining room with color: color13
    , and size: 84"w x 84"l, and price lower than 150.00 dollars
[Back to Search]
Page 1 (Total results: 50)
[Next >]
[B09KLKNRMH]
Gold Leaves Velvet Curtain Print Flannel Drape for Dining Living Room Rod Pocket
    Window Cover Light Filtering Home Decor Thermal Insulated Window Treatment 1
    Panel W39 x L78 Inch Emerald
$30.22
[B09L193GGF]
GY Blackout Window Curtain Red Stripe Wallpaper Living Room Bedroom Curtains 2 Panel
    Sets, Thermal Insulated Room Darkening Window Coverings Drapes 42W x 63L Inch
$30.87
[B09KV79XC9]
Blackout Window Curtain Panel White Dandelion 1 Pcs Curtain Coverings Rustic Wood
    Barn Retro Farmhouse Grommet Top Window Drape for Bedroom Dining Living Room 52
    x72in
```

$35.81
[B09KV7ZTN4]
Blackout Window Curtain Panel Coastal Print 1 Pcs Curtain Coverings Conch Grommet
    Top Window Drape for Bedroom Dining Living Room 52x84in
$40.5
[B09KV89NNP]
Blackout Window Curtain Panel Black and White Tulip 1 Pcs Curtain Coverings Flowers
    Blooming Grommet Top Window Drape for Bedroom Dining Living Room 52x90in
$41.82
[B09KV8FSCK]
Blackout Window Curtain Panel Independence Day 4th of July 1 Pcs Curtain Coverings
    American Flag Red White and Blue Grommet Top Window Drape for Bedroom Dining
    Living Room 52x52in
$31.17
[B09BF5LMZ8]
Marble Curtains Black and White Marble Swirl Decor Living Room Bedroom Curtains 2
    Panel Sets, Thermal Insulated Room Darkening Window Coverings Drapes 42x63inch
    (107x160cm)
$38.99
[B09KV7HT3Z]
Blackout Window Curtain Panel Valentine Day 1 Pcs Curtain Coverings Black and White
    Stripe Love Grommet Top Window Drape for Bedroom Dining Living Room 52x63in
$32.07
[B07NPDLZ3R]
HWY 50 Pink Throw Pillow Covers 18 x 18 inch, for Couch Sofa Living Room Bed,
    Decorative Embroidered Square Throw Pillow Case Cushion Cover, Fashion Geometric
    Floral, 1 Piece
$11.59
[B078NMFQ2S]
ASATEX Crushed Semi-Sheer White Color Patio Door Window Curtain Panel That's 100
    Inch Wide and 84 Inch Long Great for Sliding Doors, Extra Large Windows or
    Outdoor Coverings. RIT 100" x 84" White
$100.0

Action 2: click[B09KV79XC9]
Observation 2:
Instruction:
Find me white item window coverings for living room, dining room with color: color13
    , and size: 84"w x 84"l, and price lower than 150.00 dollars
[Back to Search] [< Prev]
size
  [52x24in] [52x36in] [52x45in]
color
  [cactuslop2663] [dandelionlop2398] [degreelop5710] [octopuslop5353]
Blackout Window Curtain Panel White Dandelion 1 Pcs Curtain Coverings Rustic Wood
    Barn Retro Farmhouse Grommet Top Window Drape for Bedroom Dining Living Room 52
    x72in
Price: $35.81 Rating: N.A.
[Description] [Features] [Reviews] [Buy Now]

Action 3: click[color13]
Observation 3: You have clicked color13.

Action 4: click[84"w x 84"l]
Observation 4: You have clicked 84"w x 84"l.

Action 5: click[Buy Now]

## B.2 ACTOR-SUMMARIZER

Instruction: Find me white item window coverings for living room, dining room with
    color: color13, and size: 84"w x 84"l, and price lower than 150.00 dollars.
[Search]

Action 1: search[white item window coverings for living room dining room]
Observation 1:
Find me white item window coverings for living room, dining room with color: color13
    , and size: 84"w x 84"l, and price lower than 150.00 dollars
[Back to Search]
Search results for keyword "white item window coverings for living room dining room

```
          ":
Page 1 (Total results: 50)
[Next >]
[B09KV79XC9]
Blackout Window Curtain Panel White Dandelion 1 Pcs Curtain Coverings Rustic Wood
     Barn Retro Farmhouse Grommet Top Window Drape for Bedroom Dining Living Room 52
     x72in
$35.81
[B09KV7ZTN4]
Blackout Window Curtain Panel Coastal Print 1 Pcs Curtain Coverings Conch Grommet
     Top Window Drape for Bedroom Dining Living Room 52x84in
$40.5
[B09KV89NNP]
Blackout Window Curtain Panel Black and White Tulip 1 Pcs Curtain Coverings Flowers
     Blooming Grommet Top Window Drape for Bedroom Dining Living Room 52x90in
$41.82
The following is in both the instruction and the description
- White
- Dining Room
- Living Room
Thw following is in the instruction, but not in description. Maybe need to check
     previous page
- color
- size
- price

Action 2: click[B09KV79XC9]
Observation 2:
[Back to Search] [< Prev]
Product info of B09KV79XC9:
size
  [52x24in] [52x36in] [52x45in]
color
  [cactuslop2663] [dandelionlop2398] [degreelop5710] [octopuslop5353]
Blackout Window Curtain Panel White Dandelion 1 Pcs Curtain Coverings Rustic Wood
     Barn Retro Farmhouse Grommet Top Window Drape for Bedroom Dining Living Room 52
     x72in
Price: $35.81 Rating: N.A.
[Description] [Features] [Reviews] [Buy Now]

Action 3: think[The item has no color13 option, maybe I should get back to search
     and reformulate the query.]
Observation 3:
OK.

Action 4: click[Back to Search]
Observation 4:
Find me white item window coverings for living room, dining room with color: color13
     , and size: 84"w x 84"l, and price lower than 150.00 dollars.
[Search]

Action 5: search[white item window coverings for living room dining room color13]
Observation 5:
Find me white item window coverings for living room, dining room with color: color13
     , and size: 84"w x 84"l, and price lower than 150.00 dollars
[Back to Search]
Search results for keyword "white item window coverings for living room dining room
     color13":
Page 1 (Total results: 50)
[Next >]
Products that does not match instruction: B09KV79XC9

[B088QVCKG3]
Animal Triple-Weaved Texture Curtain, Artistic Portrait of a White Tiger Wild Nature
      Predator Watercolor Splashes Window Covering for Living Room, Each Panel 60" W
     x 90" L Black Grey White
$124.1
[B088CNZGGQ]
Abstract Triple-Weaved Texture Curtain, Abstract Artwork Consist of Various Shapes
     and Objects with Vivid Colors Flower Window Covering for Living Room, Each Panel
```

```
          60" W x 90" L Multicolor
$124.1
[B088CY27BV]
Abstract Triple−Weaved Texture Curtain, Circles in Squares Mosaic Pattern Pastel
    Colors Modern Geometric Tile Illustration Window Covering for Living Room, Each
    Panel 60" W x 90" L Multi
$124.1
The following is in both the instruction and the description
− white item
− window coverings
− living room
− dining room
− color
− size
− price
The following is in the instruction, but not in description. Maybe need to check
    previous page
− color13

Action 6: click[B088QVCKG3]
Observation 6:
Find me white item window coverings for living room, dining room with color: color13
    , and size: 84"w x 84"l, and price lower than 150.00 dollars
[Back to Search] [< Prev]
Product info of B088QVCKG3:
size
  [84"w x 84"l]
color
  [color13]
Animal Triple−Weaved Texture Curtain, Artistic Portrait of a White Tiger Wild Nature
    Predator Watercolor Splashes Window Covering for Living Room, Each Panel 60" W
    x 90" L Black Grey White
Price: $124.1 Rating: N.A.
[Description] [Features] [Reviews] [Buy Now]

Action 7: click[color13]
Observation 7:
You have clicked color13

Action 8: click[84"w x 84"l]
Observation 8:
You have clicked 84"w x 84"l

Action 9: click[Buy Now]
```

## C ASH Prompts

We put out our prompt templates used for the SUMMARIZER and ACTOR in ASH method here. Depending
on the environment and desired behavior, we can easily adapt the SUMMARIZER prompt to guide the
ACTOR in the right direction for that particular use case.

### C.1 SUMMARIZER Prompt for CODE-DAVINCI-002

```
Action
[SEP]
None

Observation
[SEP]
WebShop
Instruction:
I want to find white blackout shades that are 66 inches in width and 66 inches in
    height. they need to be easy to install, and price lower than 110.00 dollars
[button] Search [button_]

Information
[SEP]
Let's think step by step. Based on the observation and action, what information will
    be useful for a dumb actor to buy the correct product? I need to extract useful
```

information for actor to build the correct search keyword.
info[Find me a white blackout shades with size 66" x 66". They need to be easy to
    install, and price lower than 110.00 dollars.
[Search]]

===

Action
[SEP]
search[women's tops, tees & blouses, light weight, loose fit, short sleeve, army
    green solid button, large]

Observation
[SEP]
Instruction:
Find me light weight, loose fit women's tops, tees & blouses with short sleeve with
    color: army green solid button, and size: large, and price lower than 60.00
    dollars
[button] Back to Search [button_]
Page 1 (Total results: 50)
[button] Next > [button_]
[button] B09RWQT4DC [button_]
Women's Cold Shoulder T-Shirt Tops Blouses 137 Crew-Neck Short Sleeve Printing
    Strappy Tray Fitting fit Bra Breastfeeding Men's Fixed Royal Crochet Bathing
    Casual Clothing Valentines
$19.85
[button] B09T6XK11P [button_]
Shirts for Men,Men's Casual Short Sleeve Shirts Summer Button Down Solid Color T-
    Shirt Tops Beach Loose Comfy Blouse
$3.68
[button] B09RWRSHQD [button_]
Women's Print Casual Tops Blouse T-Shirt 435 Short Sleeves Crew-Neck Letter Royal
    Cowl dot Girls' Strapless Sleeveless Fitting one Bra Adult Maternity
    Asymmetrical breastfeedi
$16.82
[button] B09RWRSQSB [button_]
Women's Soft Casual Blouse Print Tops 346 Easter Short Sleeve T Shirt ough
    Breastfeeding 2022 Lucky dot Animal Cold Fall y90k Leather Cross Swim Tankini
    dupes Leggings cro
$17.15
[button] B09T6WXZF4 [button_]
Shirts for Men,Men's Casual Short Sleeve Button Down Shirts Summer Solid Color T-
    Shirt Tops Beach Loose Comfy Blouse
$1.77
[button] B09QSTV87Q [button_]
Blouses for Women Long/Short Sleeve Solid Color Shirts Button-Down V Neck Tees with
    Pockets Tops
$12.29 to $14.28
[button] B09S3P49JZ [button_]
Loose Printed Sleeveless 266 Women's V-Neck Tank Shirt Blouse Top Block Boats
    Boutique Trendy Trending Dresser Out Strapless Embroidered Racerback Sleeved
    Short Ladie
$20.84
[button] B09NNTZCL4 [button_]
Valentines Day T Shirt for Women, Womens Tops Plus Size Cute Love Heart Graphic Tees
    Shirts Short Sleeve Tops
$1.21 to $3.21
[button] B09RWRFH91 [button_]
Women's Neck Casual Short Sleeves Tops 931 Stripes Splicing Round Strap Boys' Gym
    Waist Bodysuits Kinky Suit Tunic Leather Breastfeeding Peplum Sheer Size cor
$20.6
[button] B09S3NFW4W [button_]
Casual Easter Printing 280 Women's O-Neck Loose T-Shirt Blouse Tops Short Sleeves
    Strap Sheer Bikini Workout Cropped Winter Tunic Tube Adults See Bodysuits Teen
    Out Scrub a
$16.22

Information
[SEP]

Let's think step by step. Based on the observation and action, what information will be useful for a dumb actor to buy the correct product? First, since the actor is dumb, I should choose only about three products that is most likely to match the product and attribute specified by the instruction. I can discard the remaining products. If all products does not match the instruction for sure ( have some conflicting attributes), then I should hint the dumb actor to search a different keyword.
info[Find me light weight, loose fit women's tops, tees & blouses with short sleeve with color: army green solid button, and size: large, and price lower than 60.00 dollars.
[Back to Search]
Search results for keyword "women's tops, tees & blouses, light weight, loose fit, short sleeve, army green solid button, large":
Page 1 (Total results: 50)
[Next >]
[B09RWQT4DC]
Women's Cold Shoulder T−Shirt Tops Blouses 137 Crew−Neck Short Sleeve Printing Strappy Tray Fitting fit Bra Breastfeeding Men's Fixed Royal Crochet Bathing Casual Clothing Valentines
$19.85
[B09RWRSHQD]
Women's Print Casual Tops Blouse T−Shirt 435 Short Sleeves Crew−Neck Letter Royal Cowl dot Girls' Strapless Sleeveless Fitting one Bra Adult Maternity Asymmetrical breastfeedi
$16.82
[B09RWRSQSB]
Women's Soft Casual Blouse Print Tops 346 Easter Short Sleeve T Shirt ough Breastfeeding 2022 Lucky dot Animal Cold Fall y90k Leather Cross Swim Tankini dupes Leggings cro
$17.15]

===

Action
[SEP]
click [B078GWRC1J]

Observation
[SEP]
Instruction:
I would like a 3 ounce bottle of bright citrus deodorant for sensitive skin, and price lower than 50.00 dollars
[button] Back to Search [button_]
[button] < Prev [button_]
scent
  [button] assorted scents [button_]
  [button] bright citrus [button_]
  [button] calming lavender [button_]
  [button] ginger fresh [button_]
  [button] simply non−scents [button_]
size
  [button] travel set (4−pack) [button_]
  [button] 3 ounce (pack of 1) [button_]
  [button] 3−ounce (2−pack) [button_]
Bright Citrus Deodorant by Earth Mama | Natural and Safe for Sensitive Skin, Pregnancy and Breastfeeding, Contains Organic Calendula 3−Ounce
Price: $10.99
Rating: N.A.
[button] Description [button_]
[button] Features [button_]
[button] Reviews [button_]
[button] Buy Now [button_]

Information
[SEP]
Let's think step by step. Based on the observation and action, what information will be useful for a dumb actor to buy the correct product? First, I need to let the dumb actor know this if this is the correct product. Then, I need to filter out options that does not match the instruction to make it easier for the dumb actor to choose the correct option. Finally, if there is missing information in

the short description, I should hint the dumb actor to check the description
before buying it.
info[Find me a 3 ounce bottle of bright citrus deodorant for sensitive skin, and
price lower than 50.00 dollars
[Back to Search]
[< Prev]
Product info of B078GWRC1J:
scent
  [bright citrus]
size
  [3 ounce (pack of 1)]
  [3 ounce (pack of 2)]
Bright Citrus Deodorant by Earth Mama | Natural and Safe for Sensitive Skin,
Pregnancy and Breastfeeding, Contains Organic Calendula 3-Ounce
Price: $10.99
Rating: N.A.
[Description]
[Features]
[Reviews]
[Buy Now]]

===

Action
[SEP]
click[Description]

Observation
[SEP]
Instruction:
Find me loose fit, wide leg, quick drying, machine wash men's pants with drawstring
closure, elastic waist with color: army green, and size: x-large, and price
lower than 40.00 dollars
[button] Back to Search [button_]
[button] < Prev [button_]
Men's Outdoor Recreation Pants Convertible Quick Dry Lightweight Zip Off Outdoor
Fishing Travel Pants Men's Outdoor Casual Elastic Waist Lightweight Water
Resistant Quick Dry Cargo Fishing Hiking Pants Mens Pants Casual Workout
Drawstring Pants with Elastic Waist and Zipper Pockets Mens Pants Casual
Comfortable Workout Pants Drawstring Zipper Pockets Elastic Waist Mens Elastic
Waist Drawstring Workout Gym Joggers 3/4 Pants Summer Casual Sweat Short Pants
with Pockets Men Tactical Pants Waterproof Hiking Fishing Breathable Quick Dry
Cargo Short Pants Size:S US:4 UK:8 UE:34 Waist:64cm/25.20'' Hip:86cm/33.86''
Length:88cm/34.65'' Size:M US:6 UK:10 UE:36 Waist:68cm/26.77'' Hip:90cm/35.43''
Length:90cm/35.43''Size:L US:8 UK:12 UE:38 Waist:72cm/28.35'' Hip:9 4cm/37.01''
Length:92cm/36.22''Size:XL US:10 UK:14 UE:40 Waist:76cm/29.92'' Hip:98cm/38.58''
 Length:94cm/37.01''Size:XXL US:12 UK:16 E:42 Waist:80cm 31.50'' Hip:102cm
/40.16'' Length:96cm/37.80''

Information
[SEP]
Let's think step by step. Based on the observation and action, what information will
be useful for a dumb actor to buy the correct product? First, I need to extract
information from the description to let the dumb actor know this if this is the
correct product. Those information need to be related to the instruction, and
be concise and easy for a dumb actor to understand.
info[Find me loose fit, wide leg, quick drying, machine wash men's pants with
drawstring closure, elastic waist with color: army green, and size: x-large, and
price lower than 40.00 dollars
[Back to Search]
[< Prev]
Description of this product
The following is in both the instruction and the description
- Quick Dry
- elastic waist
- drawstring closure
Thw following is in the instruction, but not in description. Maybe need to check
previous page
- machine wash
- loose fit, wide leg

```
– color
– size
– price ]

===
```

## C.2   SUMMARIZER Prompt for GPT-3.5-TURBO

```
You are an autonomous intelligent summarizer tasked with condensing a webpage
    information by appropriately summarizing, removing and adding relevant
    information, that enables an autonomous web navigating agent to quickly and
    successfully buy a desired product.

Here's the information you will have:
1. Instruction: This is the description of the product that the agent is looking for
    .
2. Previous Action of Agent: This is the action the agent took in the previous step,
    that lead to the current webpage.
3. Observation: This is the full representation of the webpage the agent is
    currently on, after performing Previous Action, that you have to condense and
    summarize.

The agent that you want to help will be given the following information:
1. Instruction: This is the description of the product that the agent is looking for
    .
2. Observation i: This is the condensed and relevant representation of the webpage
    that you provided after the agent performed Action i-1.
3. Action i: This is the action the agent took in step i, after seeing Observation i
    provided by you.
Instruction, followed by a series of Observation and Action will be given to the
    agent to decide next optimal action.

The agent you want to help can perform the following actions:
1. search[<query >]
2. click[<button_name >]
3. think[<text >]

To be successful, it is very important for you to follow the following rules:
1. First, reason out how you want to condense the Observation for the current state
    and what information you need to add, if any.
2. Next, present one Condensed Observation in the following form: info[<Condensed
    Observation >]
3. Do not remove navigating buttons since the agent must be able to go back and
    forth between pages.
4. When there is a long list of options or products in the webpage, try to retain
    only a couple of them in the Condensed Observation based on the Instruction and
    Previous Action of Agent.
5. If "Invalid Action" is found in the Observation, add hint within info[<Condensed
    Observation >] to help the agent perform the correct action.
6. Always add (to click) after the attribute name like color, size, etc. in the
    Condensed Observation.

You are given few solved examples to help you understand your task.

### Instruction:
WebShop
Instruction:
I want to find white blackout shades that are 66 inches in width and 66 inches in
    height. they need to be easy to install, and price lower than 110.00 dollars
[button] Search [button_]

### Previous Action of Agent:
None

### Observation:
WebShop
Instruction:
I want to find white blackout shades that are 66 inches in width and 66 inches in
    height. they need to be easy to install, and price lower than 110.00 dollars
[button] Search [button_]
```

### Condensed Observation:
First, I need to remove the instruction since it is given to the agent already. Finally, I will retain the Search button so that no action is left out.
info[[Search]]

### Instruction:
WebShop
Instruction:
Find me light weight, loose fit women's tops, tees & blouses with short sleeve with color: army green solid button, and size: large, and price lower than 60.00 dollars

### Previous Action of Agent:
search[women's tops, tees & blouses, light weight, loose fit, short sleeve, army green solid button, large]

### Observation:
WebShop
Instruction:
Find me light weight, loose fit women's tops, tees & blouses with short sleeve with color: army green solid button, and size: large, and price lower than 60.00 dollars
[button] Back to Search [button_]
Page 1 (Total results: 50)
[button] Next > [button_]
[button] B09RWQT4DC [button_]
Women's Cold Shoulder T-Shirt Tops Blouses 137 Crew-Neck Short Sleeve Printing Strappy Tray Fitting fit Bra Breastfeeding Men's Fixed Royal Crochet Bathing Casual Clothing Valentines
$19.85
[button] B09T6XK11P [button_]
Shirts for Men,Men's Casual Short Sleeve Shirts Summer Button Down Solid Color T-Shirt Tops Beach Loose Comfy Blouse
$3.68
[button] B09RWRSHQD [button_]
Women's Print Casual Tops Blouse T-Shirt 435 Short Sleeves Crew-Neck Letter Royal Cowl dot Girls' Strapless Sleeveless Fitting one Bra Adult Maternity Asymmetrical breastfeedi
$16.82
[button] B09RWRSQSB [button_]
Women's Soft Casual Blouse Print Tops 346 Easter Short Sleeve T Shirt ough Breastfeeding 2022 Lucky dot Animal Cold Fall y90k Leather Cross Swim Tankini dupes Leggings cro
$17.15
[button] B09T6WXZF4 [button_]
Shirts for Men,Men's Casual Short Sleeve Button Down Shirts Summer Solid Color T-Shirt Tops Beach Loose Comfy Blouse
$1.77
[button] B09QSTV87Q [button_]
Blouses for Women Long/Short Sleeve Solid Color Shirts Button-Down V Neck Tees with Pockets Tops
$12.29 to $14.28
[button] B09S3P49JZ [button_]
Loose Printed Sleeveless 266 Women's V-Neck Tank Shirt Blouse Top Block Boats Boutique Trendy Trending Dresser Out Strapless Embroidered Racerback Sleeved Short Ladie
$20.84
[button] B09NNTZCL4 [button_]
Valentines Day T Shirt for Women, Womens Tops Plus Size Cute Love Heart Graphic Tees Shirts Short Sleeve Tops
$1.21 to $3.21
[button] B09RWRFH91 [button_]
Women's Neck Casual Short Sleeves Tops 931 Stripes Splicing Round Strap Boys' Gym Waist Bodysuits Kinky Suit Tunic Leather Breastfeeding Peplum Sheer Size cor
$20.6
[button] B09S3NFW4W [button_]
Casual Easter Printing 280 Women's O-Neck Loose T-Shirt Blouse Tops Short Sleeves Strap Sheer Bikini Workout Cropped Winter Tunic Tube Adults See Bodysuits Teen Out Scrub a

$16.22

### Condensed Observation:
First, I need to remove the instruction since it is given to the agent already. Then, I will choose only the three products closest to the description in the Instruction. I can discard the remaining products. If all products are far from the description in instruction, then I should hint the actor to go back and reformulate search query. Finally, I will retain all navigating buttons so that no action is left out.
info [[Back to Search]
Page 1 (Total results: 50)
[Next >]
[B09RWQT4DC]
Women's Cold Shoulder T-Shirt Tops Blouses 137 Crew-Neck Short Sleeve Printing Strappy Tray Fitting fit Bra Breastfeeding Men's Fixed Royal Crochet Bathing Casual Clothing Valentines
$19.85
[B09RWRSHQD]
Women's Print Casual Tops Blouse T-Shirt 435 Short Sleeves Crew-Neck Letter Royal Cowl dot Girls' Strapless Sleeveless Fitting one Bra Adult Maternity Asymmetrical breastfeedi
$16.82
[B09RWRSQSB]
Women's Soft Casual Blouse Print Tops 346 Easter Short Sleeve T Shirt ough Breastfeeding 2022 Lucky dot Animal Cold Fall y90k Leather Cross Swim Tankini dupes Leggings cro
$17.15]

### Instruction:
WebShop
Instruction:
Find me light weight, loose fit women's tops, tees & blouses with short sleeve with color: army green solid button, and size: large, and price lower than 60.00 dollars

### Previous Action of Agent:
search[women's tops, tees & blouses, light weight, loose fit, short sleeve, army green solid button, large]

### Observation:
[button] Back to Search [button_]
Page 1 (Total results: 50)
[button] Next > [button_]
[button] B09RWQT4DC [button_]
Women's Cold Shoulder T-Shirt Tops Blouses 137 Crew-Neck Short Sleeve Printing Strappy Tray Fitting fit Bra Breastfeeding Men's Fixed Royal Crochet Bathing Casual Clothing Valentines
$19.85
[button] B09T6XK11P [button_]
Shirts for Men,Men's Casual Short Sleeve Shirts Summer Button Down Solid Color T-Shirt Tops Beach Loose Comfy Blouse
$3.68
[button] B09RWRSHQD [button_]
Women's Print Casual Tops Blouse T-Shirt 435 Short Sleeves Crew-Neck Letter Royal Cowl dot Girls' Strapless Sleeveless Fitting one Bra Adult Maternity Asymmetrical breastfeedi
$16.82
Invalid Action.

### Condensed Observation:
First, Invalid Action is seen and the previous action was a search[], but there is no search button in the webpage. Hence, I should add a hint that there is no search button and the agent needs to go to search page. Then, I will retain the two products that are close to the description in the Instruction. I can discard the remaining products. Finally, I will retain all navigating buttons so that no action is left out.
info [[Back to Search]
Page 1 (Total results: 50)
[Next >]
[B09RWQT4DC]

Women's Cold Shoulder T-Shirt Tops Blouses 137 Crew-Neck Short Sleeve Printing
    Strappy Tray Fitting fit Bra Breastfeeding Men's Fixed Royal Crochet Bathing
    Casual Clothing Valentines
$19.85
[B09RWRSHQD]
Women's Print Casual Tops Blouse T-Shirt 435 Short Sleeves Crew-Neck Letter Royal
    Cowl dot Girls' Strapless Sleeveless Fitting one Bra Adult Maternity
    Asymmetrical breastfeedi
$16.82
Hint: No Search button found on this webpage, Go to search page.

### Instruction:
WebShop
Instruction:
I would like a 3 ounce bottle of bright citrus deodorant for sensitive skin, and
    price lower than 50.00 dollars

### Previous Action of Agent:
click [B078GWRC1J]

### Observation:
WebShop
Instruction:
I would like a 3 ounce bottle of bright citrus deodorant for sensitive skin, and
    price lower than 50.00 dollars
[button] Back to Search [button_]
[button] < Prev [button_]
scent
  [button] assorted scents [button_]
  [button] bright citrus [button_]
  [button] calming lavender [button_]
  [button] ginger fresh [button_]
  [button] simply non-scents [button_]
size
  [button] travel set (4-pack) [button_]
  [button] 3 ounce (pack of 1) [button_]
  [button] 3-ounce (2-pack) [button_]
Bright Citrus Deodorant by Earth Mama | Natural and Safe for Sensitive Skin,
    Pregnancy and Breastfeeding, Contains Organic Calendula 3-Ounce
Price: $10.99
Rating: N.A.
[button] Description [button_]
[button] Features [button_]
[button] Reviews [button_]
[button] Buy Now [button_]

### Condensed Observation:
First, I need to remove the instruction since it is given to the agent already. Then
    , I need to filter out options that does not match the instruction to make it
    easier for the agent to choose the correct option. If there is missing
    information in the short description, I should hint the agent to check the
    description before buying it. If there are missing options, I should hint the
    agent this might not be the correct product it searched for. Finally, I will
    retain all navigating buttons so that no action is left out.
info[[Back to Search]
[< Prev]
Product info of B078GWRC1J:
scent (to click)
 [bright citrus]
size (to click)
 [3 ounce (pack of 1)]
  [3 ounce (pack of 2)]
Bright Citrus Deodorant by Earth Mama | Natural and Safe for Sensitive Skin,
    Pregnancy and Breastfeeding, Contains Organic Calendula 3-Ounce
Price: $10.99
Rating: N.A.
[Description]
[Features]
[Reviews]
[Buy Now]]

## C.3  ACTOR Prompt for CODE-DAVINCI-002

```
Instruction : Find me a white blackout shades with size 66" x 66". They need to be
    easy to install , and price lower than 110.00 dollars .
[ Search ]

Action 1: search [66 inches in width and 66 inches in height white shades ]
Observation 1:
Find me a white blackout shades with size 66" x 66". They need to be easy to install
    , and price lower than 110.00 dollars .
[ Back to Search ]
Search results for keyword "66 inches in width and 66 inches in height white shades
    ":
Page 1 ( Total results : 50)
[ Next >]
Products clicked before : None
[ B08QGG4DY7 ]
Maxxmore Tiffany Pendant Light 2−Light 16" Width Mission Style Stained Glass Hanging
    Lamp for Kitchen Island Dining Room Foyer
$164 .0
[ B07L64LC6Y ]
Calyx Interiors Inch Snow White Cordless 2" Faux Wood Blind with Royal Valance ,
    46.5−Inch Width x 48−Inch Height
$100 .0
[ B08P24YM9D ]
Hmlinktt Couch Gray Convertible , Modern Faux Leather Futon Sofa Bed with 2 Cup
    Holders and Armrest Folding Recliner , for Living / Office Room ( Black )
$435 .0

Action 2: click [ B07L64LC6Y ]
Observation 2:
Find me a white blackout shades with size 66" x 66". They need to be easy to install
    , and price lower than 110.00 dollars .
[ Back to Search ]
[ < Prev ]
Product info of B07L64LC6Y :
size
  [19−inch width x 60−inch height ]
  [19.125−inch width x 60−inch height ]
  [21.5−inch width x 60−inch height ]
Calyx Interiors Inch Snow White Cordless 2" Faux Wood Blind with Royal Valance ,
    46.5−Inch Width x 48−Inch Height
Price : $100 .0
Rating : N.A.
[ Description ]
[ Features ]
[ Reviews ]
[ Buy Now ]

Action 3: think [ The item has no 66x66 option , maybe I should get back to search and
    reformulate the query .]
Observation 3:
OK.

Action 4: click [ Back to Search ]
Observation 4:
Find me a white blackout shades with size 66" x 66". They need to be easy to install
    , and price lower than 110.00 dollars .
[ Search ]

Action 5: search [66 x 66 blackout shades ]
Observation 5:
Find me a white blackout shades with size 66" x 66". They need to be easy to install
    , and price lower than 110.00 dollars .
[ Back to Search ]
Search results for keyword "66 x 66 blackout shades ":
Page 1 ( Total results : 50):
[ Next >]
Products clicked before : B07L64LC6Y
[ B097KXCWP7 ]
```

MiLin Blinds for Windows Cordless Blackout Cellular Shades, Blinds & Shades Room
    Darkening Honeycomb Shades Blinds Heat Insulation UV Protection for Home Kitchen
     Bedroom Office — White 28" W x 36" H
$78.49
[B082GGGM2Y]
SUN AWAY Portable Blackout Curtain with Window Suction Cups (66" Long x 51" Wide)
    Temporary Room Darkening Shades for Baby Nursery, Bedroom, Dorm Room — Window
    Cover for Travel
$42.99
[B09GP34TDR]
Ejiawj Painted Total Blackout Curtains for Bedroom Plant Comics Farm House Curtains
    66 x 72 inch Grommet Room Blackout Curtains Heat Insulation Noise Reduction and
    Blackout Curtains Suitable for Livi
$100.0

Action 6: click[B097KXCWP7]
Observation 6:
Find me a white blackout shades with size 66" x 66". They need to be easy to install
    , and price lower than 110.00 dollars.
[Back to Search]
Product info of B097KXCWP7:
[< Prev]
size
  [66"w x 56"h]
  [66"w x 60"h]
  [66"w x 66"h]
color
  [cordless bottom up—blackout—creamy]
  [cordless bottom up—blackout—gray]
  [cordless bottom up—blackout—white]
MiLin Blinds for Windows Cordless Blackout Cellular Shades, Blinds & Shades Room
    Darkening Honeycomb Shades Blinds Heat Insulation UV Protection for Home Kitchen
     Bedroom Office — White 28" W x 36" H
Price: $78.49
Rating: N.A.
[Description]
[Features]
[Reviews]
[Buy Now]

Action 7: think[For white blackout shades that are 66 inches in width and 66 inches
    in height, the item has options '66"w x 66"h' and 'cordless bottom up—blackout—
    white', so I should buy this item.]
Observation 7:
OK.

Action 8: click[66"w x 66"h]
Observation 8:
You have clicked 66"w x 66"h.

Action 9: click[cordless bottom up—blackout—white]
Observation 9:
You have clicked cordless bottom up—blackout—white.

Action 10: click[Buy Now]

===

Instruction: Find me a 3 ounce bottle of bright citrus deodorant for sensitive skin,
    and price lower than 50.00 dollars
[Search]

Action 1: search[3 ounce bright citrus deodorant sensitive skin]
Observation 1:
Find me a 3 ounce bottle of bright citrus deodorant for sensitive skin, and price
    lower than 50.00 dollars
[Back to Search]
Search results for keyword "3 ounce bright citrus deodorant sensitive skin":
Page 1 (Total results: 50)
[Next >]

```
Products clicked before: None
[B078GWRC1J]
Bright Citrus Deodorant by Earth Mama | Natural and Safe for Sensitive Skin,
    Pregnancy and Breastfeeding, Contains Organic Calendula 3-Ounce
$10.99
[B078GTKVXY]
Ginger Fresh Deodorant by Earth Mama | Natural and Safe for Sensitive Skin,
    Pregnancy and Breastfeeding, Contains Organic Calendula 3-Ounce
$10.99
[B08KBVJ4XN]
Barrel and Oak - Aluminum-Free Deodorant, Deodorant for Men, Essential Oil-Based
    Scent, 24-Hour Odor Protection, Cedar & Patchouli Blend, Gentle on Sensitive
    Skin (Mountain Sage, 2.7 oz, 2-Pack)
$15.95

Action 2: click[B078GWRC1J]
Observation 2:
Find me a 3 ounce bottle of bright citrus deodorant for sensitive skin, and price
    lower than 50.00 dollars
[Back to Search]
[< Prev]
Product info of B078GWRC1J:
scent
  [bright citrus]
size
  [3 ounce (pack of 1)]
  [3 ounce (pack of 2)]
Bright Citrus Deodorant by Earth Mama | Natural and Safe for Sensitive Skin,
    Pregnancy and Breastfeeding, Contains Organic Calendula 3-Ounce
Price: $10.99
Rating: N.A.
[Description]
[Features]
[Reviews]
[Buy Now]

Action 3: click[bright citrus]
Observation 3:
You have clicked bright citrus.

Action 4: click[3 ounce (pack of 1)]
Observation 4:
You have clicked 3 ounce (pack of 1).

Action 5: click[Buy Now]

===
```

## C.4  ACTOR Prompt for GPT-3.5-TURBO

```
You are an autonomous intelligent agent tasked with navigating a shopping website
    called "WebShop" to buy a desired product. These tasks will be accomplished
    through the use of specific actions you can issue.

Here's the information you'll have:
1. Instruction: This is the description of the product you are looking for.
2. Observation i: This is a simplified representation of the webpage you face after
    performing Action i-1, providing key information and buttons you can click on.
3. Action i: This is the action you took in step i, after seeing Observation i.
Instruction will be found at the beginning of the task, followed by a series of
    Observation and Action. {}

You can perform the following actions:
1. search[<query>] - search for products described by <query>.
2. click[<button_name>] - click on a button present in webpage where <button_name>
    is the name of the button given as [button_name] in Observation.
3. think[<text>] - think and reason about what needs to be done given the history of
     observations and actions. Your next action will be based on this thought.

To be successful, it is very important to follow the following rules:
```

1. Issue Action in the correct format, ensuring it is supported by the Observation, and issue only one Action at a time.
2. Do not repeat Action like clicking the same product or searching the same query.
3. If "Invalid Action" is found in the Observation, the Action performed is not supported by the webpage. Try a different Action that is supported by the webpage or navigate to the webpage that supports that Action.
4. Do not perform think[<text>] Action if "Thought Through" is present in the Observation.
5. Do not always look for exact string matches between Instruction and the product description in the webpage. Look if the meaning of the product description matches the Instruction.
6. Follow Hint in Observation carefully.

You are given few solved examples to help you understand your task.

### Instruction:
WebShop
Instruction:
i want to find white blackout shades that are 66 inches in width and 66 inches in height. they need to be easy to install, and price lower than 110.00 dollars
[Search]

### Observation 1:
WebShop
Instruction:
i want to find white blackout shades that are 66 inches in width and 66 inches in height. they need to be easy to install, and price lower than 110.00 dollars
[Search]

### Action 1:
search[66 inches width and 66 inches height white shades]

### Observation 2:
[Back to Search]
Page 1 (Total results: 50)
[Next >]
[B08QGG4DY7]
Maxxmore Tiffany Pendant Light 2-Light 16" Width Mission Style Stained Glass Hanging Lamp for Kitchen Island Dining Room Foyer
$164.0
[B07L64LC6Y]
Calyx Interiors Inch Snow White Cordless 2" Faux Wood Blind with Royal Valance, 46.5-Inch Width x 48-Inch Height
$100.0
[B08P24YM9D]
Hmlinktt Couch Gray Convertible, Modern Faux Leather Futon Sofa Bed with 2 Cup Holders and Armrest Folding Recliner, for Living/Office Room(Black)
$435.0

### Action 2:
click[B07L64LC6Y]

### Observation 3:
[Back to Search]
[< Prev]
size [19-inch width x 60-inch height][19.125-inch width x 60-inch height][21.5-inch width x 60-inch height]
Calyx Interiors Inch Snow White Cordless 2" Faux Wood Blind with Royal Valance, 46.5-Inch Width x 48-Inch Height
Price: $100.0
Rating: N.A.
[Description]
[Features]
[Reviews]
[Buy Now]

### Action 3:
think[The item has no 66x66 option, maybe I should get back to search and reformulate the query.]

### Observation 4:
Thought Through.

### Action 4:
click[Back to Search]

### Observation 5:
WebShop
Instruction:
i want to find white blackout shades that are 66 inches in width and 66 inches in
    height. they need to be easy to install, and price lower than 110.00 dollars
[Search]

### Action 5:
search[66 x 66 blackout shades]

### Observation 6:
[Back to Search]
Page 1 (Total results: 50)
[Next >]
[B097KXCWP7]
MiLin Blinds for Windows Cordless Blackout Cellular Shades, Blinds & Shades Room
    Darkening Honeycomb Shades Blinds Heat Insulation UV Protection for Home Kitchen
     Bedroom Office – White 28" W x 36" H
$78.49
[B082GGGM2Y]
SUN AWAY Portable Blackout Curtain with Window Suction Cups (66" Long x 51" Wide)
    Temporary Room Darkening Shades for Baby Nursery, Bedroom, Dorm Room – Window
    Cover for Travel
$42.99
[B09GP34TDR]
Ejiawj Painted Total Blackout Curtains for Bedroom Plant Comics Farm House Curtains
    66 x 72 inch Grommet Room Blackout Curtains Heat Insulation Noise Reduction and
    Blackout Curtains Suitable for Livi
$100.0

### Action 6:
click[B097KXCWP7]

### Observation 7:
[Back to Search]
[< Prev]
size
 [66"w x 56"h][66"w x 60"h][66"w x 66"h][custom size]
color
 [cordless bottom up-blackout-creamy][cordless bottom up-blackout-gray][cordless
    bottom up-blackout-white]
MiLin Blinds for Windows Cordless Blackout Cellular Shades, Blinds & Shades Room
    Darkening Honeycomb Shades Blinds Heat Insulation UV Protection for Home Kitchen
     Bedroom Office – White 28" W x 36" H
Price: $78.49
Rating: N.A.
[Description]
[Features]
[Reviews]
[Buy Now]

### Action 7:
think[For white blackout shades that are 66 inches in width and 66 inches in height,
    the item has options '66"w x 66"h' and 'cordless bottom up-blackout-white', so
    I should buy this item.]

### Observation 8:
Thought Through.

### Action 8:
click[66"w x 66"h]

### Observation 9:
You have clicked 66"w x 66"h.

### Action 9:
click [ cordless bottom up−blackout−white ]

### Observation 10:
You have clicked cordless bottom up−blackout−white.

### Action 10:
click [ Buy Now ]

### Instruction:
WebShop
Instruction:
i would like a 3 ounce bottle of bright citrus deodorant for sensitive skin, and
    price lower than 50.00 dollars
[ Search ]

### Observation 1:
WebShop
Instruction:
i would like a 3 ounce bottle of bright citrus deodorant for sensitive skin, and
    price lower than 50.00 dollars
[ Search ]

### Action 1:
search [3 ounce bright citrus deodorant sensitive skin]

### Observation 2:
[ Back to Search ]
Page 1 (Total results: 50)
[ Next > ]
[B078GWRC1J]
Bright Citrus Deodorant by Earth Mama | Natural and Safe for Sensitive Skin,
    Pregnancy and Breastfeeding, Contains Organic Calendula 3−Ounce
$10.99
[B078GTKVXY]
Ginger Fresh Deodorant by Earth Mama | Natural and Safe for Sensitive Skin,
    Pregnancy and Breastfeeding, Contains Organic Calendula 3−Ounce
$10.99
[B08KBVJ4XN]
Barrel and Oak − Aluminum−Free Deodorant, Deodorant for Men, Essential Oil−Based
    Scent, 24−Hour Odor Protection, Cedar & Patchouli Blend, Gentle on Sensitive
    Skin (Mountain Sage, 2.7 oz, 2−Pack)
$15.95

### Action 2:
click [B078GWRC1J]

### Observation 3:
[ Back to Search ]
[ < Prev ]
scent
 [ assorted scents ][ bright citrus ][ calming lavender ][ ginger fresh ][ simply non−scents ]
size
 [ travel set (4−pack) ][3 ounce (pack of 1) ][3−ounce (2−pack) ]
Bright Citrus Deodorant by Earth Mama | Natural and Safe for Sensitive Skin,
    Pregnancy and Breastfeeding, Contains Organic Calendula 3−Ounce
Price: $10.99
Rating: N.A.
[ Description ]
[ Features ]
[ Reviews ]
[ Buy Now ]

### Action 3:
click [ bright citrus ]

### Observation 4:
You have clicked bright citrus.

```
### Action 4:
click[3 ounce (pack of 1)]

### Observation 5:
You have clicked 3 ounce (pack of 1).

### Action 5:
click[Buy Now]
```

## D SUMMARIZER Output Example

Here we show one example, where the SUMMARIZER eliminates irrelevant information by picking the top three relevant products in the search result.

```
raw observation (input of SUMMARIZER)
---
Instruction:
Find me loose fit, wide leg, quick drying, machine wash men's pants with drawstring
    closure, elastic waist with color: army green, and size: x-large, and price
    lower than 40.00 dollars
[button] Back to Search [button_]
Page 1 (Total results: 50)
[button] Next > [button_]
[button] B09Q6KK8WC [button_]
Mens Classic Fit Ripstop Cargo Shorts Big and Tall Outdoor Casual Quick Dry Hiking
    Tactical Workout Short with Pockets
$18.29
[button] B09C393J7W [button_]
HUANLE Men Fitness Sports Yoga Pants Men's Compression Pants Quick Dry Capri
    Athletic Leggings Workout Tights
$11.99 to $12.79
[button] B09RB3H4KF [button_]
Women's Loose Fit Elastic Waist Drawstring Lace Shorts Plus Size Summer Beach Casual
     Yoga Lounge Shorts Pants S-5XL
$9.99
[button] B09RZK8742 [button_]
Merthy Mens Cotton Linen Drawstring Pants Casual Classic Fit 11 Inch Inseam Elastic
    Waist Shorts Summer Beach Yoga Short
$6.14
[button] B08ZRT9N1T [button_]
Mordenmiss Women's Baggy Drawstring Pants Wide Leg Pants Casual Elastic Waist
    Trousers
$25.99 to $33.99
[button] B09SZFTFJ4 [button_]
Merthy Linen Casual Classic Fit Shorts for Mens Drawstring Summer Beach Short
    Elastic Waist Sleep Pajama Pants with Pockets
$4.88
[button] B09SHTBSTG [button_]
Merthy Mens Lounge Shorts Drawstring Elastic Waist Tactical Workout Cargo Short
    Loose Fit Gym Fishing Hiking Palazzo Pants
$4.99
[button] B094975Y71 [button_]
Wide Leg Pants for Women Elastic Waist Casual Capris Crop Pull On Pants Dandelion
    Harem Trousers Cropped Pants with Pockets
$5.99 to $16.88
[button] B09T3LT1TN [button_]
Merthy Summer Casual Rompers for Women Deep V Neck Spaghetti Strap Drawstring Waist
    Terry Jumpsuits Overalls with Pockets
$11.99
[button] B09R9ZV4VY [button_]
melupa Summer Pants for Women Casual Pockets Cotton Linen Wide Leg Trousers
    Drawstring Elastic Waist Jogging Yoga Sweatpants
$14.99
---

Summarizer output
---
Find me loose fit, wide leg, quick drying, machine wash men's pants with drawstring
    closure, elastic waist with color: army green, and size: x-large, and price
    lower than 40.00 dollars
```

```
[Back to Search]
Search results for keyword "loose fit wide leg quick drying machine wash men's pants
    drawstring closure elastic waist army green x-large":
Page 1 (Total results: 50)
[Next >]
[B09Q6KK8WC]
Mens Classic Fit Ripstop Cargo Shorts Big and Tall Outdoor Casual Quick Dry Hiking
    Tactical Workout Short with Pockets
$18.29
[B09RZK8742]
Merthy Mens Cotton Linen Drawstring Pants Casual Classic Fit 11 Inch Inseam Elastic
    Waist Shorts Summer Beach Yoga Short
$6.14
[B09SZFTFJ4]
Merthy Linen Casual Classic Fit Shorts for Mens Drawstring Summer Beach Short
    Elastic Waist Sleep Pajama Pants with Pockets
$4.88
```

# E  SUMMARIZER Output Analysis

## E.1  SUMMARIZER Failure Example 1

The SUMMARIZER fail when the raw observation is too long. It simply repeats many of the options without effective summarization.

```
Raw Observation:
---
Instruction:
Find me easy care, moisture wicking, machine wash men's pants with polyester cotton
    with color: black, and size: 29w x 30l, and price lower than 50.00 dollars
[button] Back to Search [button_]
[button] < Prev [button_]
color
  [button] olive-pleated front [button_]
  [button] navy-pleated front [button_]
  [button] brown-pleated front [button_]
  [button] stone-pleated front [button_]
  [button] black-pleated front [button_]
  [button] khaki-flat front [button_]
  [button] black-flat front [button_]
  [button] khaki-pleated front [button_]
  [button] charcoal-flat front [button_]
  [button] white-pleated front [button_]
  [button] black - skinny fit-flat front [button_]
  [button] navy-flat front [button_]
  [button] olive-flat front [button_]
  [button] white-flat front [button_]
size
  [button] 28w x 28l [button_]
  [button] 28w x 30l [button_]
  [button] 28w x 32l [button_]
  [button] 29w x 28l [button_]
  [button] 29w x 29l [button_]
  [button] 29w x 30l [button_]
  [button] 29w x 32l [button_]
  [button] 30w x 29l [button_]
  [button] 30w x 30l [button_]
  [button] 30w x 32l [button_]
  [button] 30w x 34l [button_]
  [button] 31w x 29l [button_]
  [button] 31w x 30l [button_]
  [button] 31w x 32l [button_]
  [button] 32w x 28l [button_]
  [button] 32w x 32l [button_]
  [button] 32w x 34l [button_]
  [button] 33w x 28l [button_]
  [button] 33w x 30l [button_]
  [button] 33w x 32l [button_]
  [button] 34w x 30l [button_]
```

[button] 34w x 32l [button_]
[button] 36w x 30l [button_]
[button] 36w x 32l [button_]
[button] 36w x 36l [button_]
[button] 38w x 30l [button_]
[button] 38w x 32l [button_]
[button] 40w x 30l [button_]
[button] 40w x 32l [button_]
[button] 40w x 34l [button_]
[button] 42w x 30l [button_]
[button] 44w x 30l [button_]
[button] 44w x 32l [button_]
[button] 44w x 34l [button_]
[button] 46w x 30l [button_]
[button] 46w x 32l [button_]
[button] 48w x 30l [button_]
[button] 48w x 32l [button_]
[button] 48w x 36l [button_]
[button] 50w x 30l [button_]
[button] 50w x 32l [button_]
[button] 52w x 30l [button_]
[button] 52w x 32l [button_]
[button] 54w x 30l [button_]
[button] 54w x 32l [button_]
[button] 56w x 30l [button_]
[button] 56w x 32l [button_]
[button] 58w x 30l [button_]
[button] 58w x 32l [button_]
[button] 60w x 30l [button_]
[button] 60w x 32l [button_]
AKA Men's Wrinkle Free Cotton Twill - Traditional Fit Slacks Chino Straight-Legs
    Casual Pants
Price: $28.61 to $47.99
Rating: N.A.
[button] Description [button_]
[button] Features [button_]
[button] Reviews [button_]
[button] Buy Now [button_]
---

Summarizer output
---
Find me easy care, moisture wicking, machine wash men's pants with polyester cotton
    with color: black, and size: 29w x 30l, and price lower than 50.00 dollars
[Back to Search]
[< Prev]
Product info of B01N45QG95:
color
  [black-pleated front]
size
  [29w x 30l]
  [29w x 32l]
  [30w x 30l]
  [30w x 32l]
  [30w x 34l]
  [31w x 30l]
  [31w x 32l]
  [32w x 30l]
  [32w x 32l]
  [32w x 34l]
  [33w x 30l]
  [33w x 32l]
  [34w x 30l]
  [34w x 32l]
  [36w x 30l]
  [36w x 32l]
  [36w x 36l]
  [38w x 30l]
  [38w x 32l]
  [40w x 30l]

```
    [40w x 32l]
    [40w x 34l]
    [42w x 30l]
    [44w x 30l]
    [44w x 32l]
    [44w x 34l]
    [46w x 30l]
    [46w x 32l]
    [48w x 30l]
    [48w x 32l]
    [48w x 36l]
    [50w x 30l]
    [50w x 32l]
    [52w x 30l]
    [52w x 32l]
    [54w x 30l]
    [54w x 32l]
    [56w x 30l]
    [56w x 32l]
    [58w x 30l]
    [58w x 32l]
    [60w x 30l]
    [60w x 32l]
AKA Men's Wrinkle Free Cotton Twill - Traditional Fit Slacks Ch
---

Expected Output
---
Find me easy care, moisture wicking, machine wash men's pants with polyester cotton
    with color: black, and size: 29w x 30l, and price lower than 50.00 dollars
[Back to Search]
[< Prev]
Product info of B01N45QG95:
color
  [black-pleated front]
size
  [29w x 30l]
AKA Men's Wrinkle Free Cotton Twill - Traditional Fit Slacks Chino Straight-Legs
    Casual Pants
Price: $28.61 to $47.99
Rating: N.A.
[Description]
[Features]
[Reviews]
[Buy Now]
---
```

## E.2 SUMMARIZER Failure Example 2

SUMMARIZER fails because of hallucination. It hallucinates an non-existing size option '[34w x 34l]'
which is required by the instruction.

```
raw observation (input of SUMMARIZER)
---
Instruction:
Find me low rise, straight leg, slim fit, machine wash men's jeans with imported
    zipper, polyester spandex with color: jet set blue, and size: 34w x 34l, and
    price lower than 90.00 dollars
[button] Back to Search [button_]
[button] < Prev [button_]
size
  [button] 25w x 32l [button_]
  [button] 30 [button_]
  [button] 30w x 38l [button_]
  [button] 32 [button_]
  [button] 32w x 30l [button_]
  [button] 32w x 34l [button_]
  [button] 32w x 36l [button_]
  [button] 32w x 38l [button_]
```

```
  [button] 33 [button_]
  [button] 33w x 30l [button_]
  [button] 33w x 38l [button_]
  [button] 34 [button_]
  [button] 34w x 30l [button_]
  [button] 34w x 36l [button_]
  [button] 34w x 38l [button_]
  [button] 35w x 36l [button_]
  [button] 36 [button_]
  [button] 36w x 30l [button_]
  [button] 38 [button_]
  [button] 40w x 32l [button_]
  [button] 40w x 34l [button_]
  [button] 44w x 36l [button_]
ARIAT Men's M4 Low Rise Stretch Shoreway Stackable Straight Leg Jean
Price: $79.99 to $166.75
Rating: N.A.
[button] Description [button_]
[button] Features [button_]
[button] Reviews [button_]
[button] Buy Now [button_]
---

Summarizer output
---
Find me low rise, straight leg, slim fit, machine wash men's jeans with imported
    zipper, polyester spandex with color: jet set blue, and size: 34w x 34l, and
    price lower than 90.00 dollars
[Back to Search]
[< Prev]
Product info of B084G9PT3P:
size
  [34w x 34l]
ARIAT Men's M4 Low Rise Stretch Shoreway Stackable Straight Leg Jean
Price: $79.99 to $166.75
Rating: N.A.
[Description]
[Features]
[Reviews]
[Buy Now]
---
```

## F   Environment Setup Code

```python
env = WebAgentTextEnv( # From WebShop repo
    observation_mode='text_rich',
    render=False,
    num_products=None, # full product space
  )

for seed in range(501, 1000):
    random.seed(seed)
    session = "".join(random.choices(string.ascii_lowercase, k=10))
    (ob, _) = env.reset(session=session)
    # test LLM agent on this session
```

