# OpenReview forum: "Hierarchical Prompting Assists Large Language Model on Web Navigation"
_EMNLP/2023/Conference — EMNLP 2023 Findings_

### Official Review · Reviewer_KCDF · 2023-07-30

**Soundness:** 2

**Excitement:**

2: Mediocre: This paper makes marginal contributions (vs non-contemporaneous work), so I would rather not see it in the conference.

**Paper Topic And Main Contributions:**

The paper introduces a novel approach called Actor-Summarizer-Hierarchical (ASH) prompt for Web Navigation. Unlike previous prompting methods, the paper argues that filtering out irrelevant information from web observations before inputting them into Language Model (LLMs) is essential. The proposed ASH prompt consists of two components: SUMMARIZER and ACTOR prompts.

The SUMMARIZER prompt is designed to eliminate irrelevant details from the web observations, and the processed data is then fed into LLMs with the ACTOR prompt, which focuses on predicting the next step. ASH achieved a task success rate that outperforms the previous state-of-the-art prompting mechanism using the same LLM by 6.2%. These results demonstrate the potential of our approach in interactive decision-making tasks involving long observation traces.

**Questions For The Authors:**

1. Regarding the statement on line 184, the paper lacks clarity on what is meant by the "simpler version of dataset taken from Yao et al." It remains unclear if this dataset has been utilized in other research papers as well. Further elaboration and a reference to the original source would be beneficial to better understand its origin and potential usage in related studies.

2. In Figure 4, how many examples, featuring more than 11 steps, are the same between REACT and ASH?

**Reasons To Accept:**

1. The presented SUMMARIZER prompt, which effectively filters out irrelevant information from observations, demonstrates encouraging outcomes in the context of web navigation.

**Reasons To Reject:**

1. The paper lacks specific details on how SUMMARIZER effectively eliminates irrelevant information. Particularly, it does not clearly define what qualifies as irrelevant information that should be avoided when providing input to the LLM. Additionally, there is a lack of an error study on SUMMARIZER, which could help determine whether there is a possibility of crucial information being mistakenly removed by the prompt.

2. The paper's failure to provide comprehensive details about the baseline models for comparison, such as REACT, is a notable limitation. The differences between ASH and REACT are not highlighted, making it challenging to discern the most crucial components of ASH that contribute to its superiority over existing approaches.

3. The evaluation in the paper is not sufficiently comprehensive and lacks transparency regarding the experiment setup. For instance, there is no mention of the number of different sets of in-content examples used in the experiments. Additionally, the paper does not explore the effects of varying the number of In-Context Examples. Moreover, the evaluation relies solely on one dataset, which may limit the generalizability of the results.

**Reproducibility:**

3: Could reproduce the results with some difficulty. The settings of parameters are underspecified or subjectively determined; the training/evaluation data are not widely available.

**Reviewer Confidence:**

3: Pretty sure, but there's a chance I missed something. Although I have a good feel for this area in general, I did not carefully check the paper's details, e.g., the math, experimental design, or novelty.

---

> ### Author Rebuttal · Authors · 2023-08-29
>
> Thank you for your thorough reviews and valuable feedback!
>
>
>
> * How SUMMARIZER works
>     * In our work how SUMMARIZER identifies and removes irrelevant information is guided by the provided in-context examples. SUMMARIZER will determine how useful an information is by considering how helpful or relevant it is for an actor to achieve the goal.
>     * Here we show one example ([https://pastebin.com/tRpTFY6k](https://pastebin.com/tRpTFY6k) ), where the SUMMARIZER eliminates irrelevant information by picking the top three relevant products in the search result. This effectively makes the observation space of ACTOR smaller, which leads to better performance.
> * Error analysis for SUMMARIZER
>     * Since we implemented our SUMMARIZER by prompting LLM, common error analysis of LLM applies here. For example, SUMMARIZER fails to summarize properly when raw observation is really long ([https://pastebin.com/CdLRC1VR](https://pastebin.com/CdLRC1VR)); SUMMARIZER hallucinates ([https://pastebin.com/bWfzSuG1](https://pastebin.com/bWfzSuG1)), adding information that does not exist in the raw observation. Besides those, ASH prompting also provides scope for improving the quality of SUMMARIZER. For example, this example ([https://pastebin.com/fz5kGHEk](https://pastebin.com/fz5kGHEk)) shows that SUMMARIZER can hint ACTOR on how to recover when it performs an Invalid Action or what it needs to do after a Think action. Such provisions can allow efficient recovery of agents compared to methods which rely solely on ACTOR prompting.
> * Difference between REACT and ASH
>     * The difference between REACT and ASH is that ASH makes two passes over the observation, removing irrelevant information in the first pass and making the final decision in the second pass. Notice that this process can be generalized to multi-pass, where we have multiple SUMMARIZERs and each of them has its own job of attending to specific information or usecases.
>     * This difference leads to a different way of processing raw observation from the WebShop environment, which is briefly mentioned in the footnote of page 3. Usually the search listing page will have 10 top search results. How REACT processes it is to just pick the top 3 and discard the remaining result, while what ASH does is to pass all 10 search results to SUMMARIZER, and the SUMMARIZER will pick the 3 most relevant search results. Due to limitations of the search engine, the top 3 of the search results might not always be the most relevant, as shown in the example provided above. In addition, longer observations have a higher chance of driving agents towards undesirable trajectories. Derivative works of REACT also follows the same setup.
>     * As for the experimental setup, we retain the complexity of the webpages differing from the ReAct paper, without any manual filtering.
>     * The main advantage of ASH is the SUMMARIZER being orthogonal to the development of better ACTOR prompts, which the existing works have majorly focused on. Infact, our main method uses ReAct based ACTOR prompting conjugated with the SUMMARIZER. We have also demonstrated using a Act based ACTOR prompting and added SUMMARIZER in the loop. In future, we expect that more sophisticated ACTOR and SUMMARIZER prompts can be individually developed and combined to achieve superlative performances. We put out our novel idea as a short paper due to this focused contribution.
> * Evaluation details and clarification for dataset setup
>     * We apologize for the missing experiments and evaluations details.The prompts we used for SUMMARIZER and ACTOR in Appendix D. We provide 4 in-context examples for the SUMMARIZER and provide 2 in-context examples for the ACTOR. Note that in Appendix C we also provided an example trajectory for ASH and an example trajectory for REACT.
>     * For the evaluation environment, we generate 500 session strings controlled by fixed seeds, as denoted below. Each session string corresponds to a specific instruction i.e. a specific task.
>
>              env = WebAgentTextEnv( # From WebShop repo
>                 observation_mode='text_rich',
>                 render=False,
>                 num_products=None, # full product space
>              )
>
>              for seed in range(501, 1000):
>                  random.seed(seed)
>                  session = "".join(random.choices(string.ascii_lowercase, k=10))
>                  (ob, _) = env.reset(session=session)
>                  # test LLM agent on this session
>
> * For score calculation, there are two possibilities
> 1. The trajectory ends normally i.e. it ends with the agent choosing to buy some product. In this case the environment will calculate a score between zero and one, representing the percentage of the required attributes that are satisfied by the product. When this score is 1.0, we view this trajectory as a “success” trajectory. When this score is 0.0, we view this trajectory as a “failure” trajectory”
> 2. The trajectory ends abnormally. A trajectory will be terminated if (1) it produces 5 consecutive invalid actions (2) the trajectory length exceeds 20. In this case we will assign a score of 0.0 to this trajectory.
>
> We will succinctly add the above clarifications and information to the revised paper, as necessary, for better clarity and reproducibility.
>
>
>
> * “In Figure 4, how many examples, featuring more than 11 steps, are the same between REACT and ASH?”
>     * Two methods have an (almost) non-overlapping set of examples that features more than 11 steps. Note that ASH has way more examples that feature more than 11 steps compared to REACT.
>
> The work is aimed at complex observation space sequential tasks, like web navigation, as the title suggests and at the time of writing the paper, this was the only relevant dataset to the best of our knowledge. However, we acknowledge that the work demonstrates the method’s efficacy only on two older LLMs (including the text-davinci-002 GPT3 in Appendix B). To this front, we test our method’s generalizability to newer, stronger LLMs like ChatGPT (gpt-3.5-turbo). We choose this LLM to balance the strength and cost of running the experiment. We also chose the 4K token limit variant to test generalization to smaller token limit LLMs. While this initial analysis was run on a smaller product space with 1000 products and using 100 test samples, we will adopt the same sophisticated settings as the main results in the paper, and add it to the revised version of the paper along with the respective error analysis.
>
> As shown in the table below, ASH surpasses ReAct’s performance on ChatGPT, mirroring the same trend as when we were using the relatively weaker Codex models or GPT3. While the major merit of ASH in this setting has been its ability to condense observation allowing longer trajectories on the smaller token limit models, its ability to help agent recover from errors and align to the optimal behavior still holds as we observed on few sample test trajectories. The pronounced improvement of success rate with ASH can also be attributed to the nature of ChatGPT agent: the agent attempts to buy the exact product and does not frequently settle with suboptimal products, compared to the older LLMs. We will also release the code and prompts for the chatgpt experiments in the same github repository as our main code.
>
>
> | Model	| ASH Avg Score |	ASH Success % |	ReAct Avg Score |	ReAct Success % |
> | ----------| ------------------ | ------------------ | ------------------ | ------------------ |
> | Codex (code-davinci-002) on 50 samples |	60.9	| 23.1 |	50.2	| 15.3 |
> | GPT3 (text-davinci-002) on 50 samples |	62.6	| 28.8 |	61.3	| 23.1 |
> | ChatGPT (gpt-3.5-turbo) on 100 samples |	59.9	| 46.0 |	17.8	| 10.0 |

---

### Official Review · Reviewer_g69r · 2023-08-04

**Soundness:** 3

**Excitement:**

3: Ambivalent: It has merits (e.g., it reports state-of-the-art results, the idea is nice), but there are key weaknesses (e.g., it describes incremental work), and it can significantly benefit from another round of revision. However, I won't object to accepting it if my co-reviewers champion it.

**Paper Topic And Main Contributions:**

This manuscript proposes a method called Hierarchical Hint Assistant for optimizing the performance of LLMs on web navigation tasks. The effect of the proposed method is verified by experiments on the Webshop dataset.

**Questions For The Authors:**

Is it possible to test the proposed method on the text-davinci-003-based model to verify the effectiveness of the proposed method?

**Reasons To Accept:**

The method proposed in this paper can be used to deal with the optimization of LLMs in complex interaction scenarios. It has certain reference significance for research in this field. The structure of the article is good, and the expression is clear and easy to read.

**Reasons To Reject:**

The experiments of this work are not very sufficient, and it has not been fully verified on the current representative LLMs, and there is also a lack of corresponding ablation experiments.

**Reproducibility:**

4: Could mostly reproduce the results, but there may be some variation because of sample variance or minor variations in their interpretation of the protocol or method.

**Reviewer Confidence:**

3: Pretty sure, but there's a chance I missed something. Although I have a good feel for this area in general, I did not carefully check the paper's details, e.g., the math, experimental design, or novelty.

---

> ### Author Rebuttal · Authors · 2023-08-29
>
> Thank you for your thorough reviews and valuable feedback. We understand the concern of testing only two older LLMs in the paper (the result for GPT-3 is provided in Appendix B), and have run our prompting method ASH on another new, chat-based LLM: ChatGPT (gpt-3.5-turbo). We choose this LLM to balance the strength and cost of running the experiment. We also chose the 4K token limit variant to test generalization to smaller token limit LLMs. While this initial analysis was run on a smaller product space with 1000 products and using 100 test samples, we will adopt the same sophisticated settings as the main results in the paper, and add it to the revised version of the paper along with the respective error analysis.
>
> As shown in the table below, ASH surpasses ReAct’s performance on ChatGPT, mirroring the same trend as when we were using the relatively weaker Codex models or GPT3. While the major merit of ASH in this setting has been its ability to condense observation allowing longer trajectories on the smaller token limit models, its ability to help agent recover from errors and align to the optimal behavior still holds as observed on few sample test trajectories. The pronounced performance improvement with ASH can be attributed to the nature of ChatGPT agent: the agent attempts to buy the exact product and does not frequently settle with suboptimal products, compared to the older LLMs. In fact when we provide examples of successful trajectories this behaviour is enhanced, as seen by the higher average score when ACTOR is zero-shot prompted. We will also release the code and prompts for the chatgpt experiments in the same github repository as our main code.
>
> One important ablation study that we wish to add to our revised paper is varying the numbers of in-context examples in the ACTOR prompt to see how that affects the relative performance of ASH vs ReAct. While we look forward to add an in-detail analysis on this in the revised version, we report the performance on two points now (current version that uses a 2-shot ACTOR and a zero-shot ACTOR) here. in the and found that when we provide instruction without in-context examples, ASH performs more robust then REACT. Furthermore, when combining the summarizer and actor to one prompt, the result shows that ASH’s two-step prompting method still performs better.
>
> | Model                                                             | **ASH**  Avg Score | **ASH** Success %     | **ReAct**  Avg Score | **ReAct** Success % |
> | ----------------------------------------------------------------- | --------- | --------- | --------- | --------- |
> | Codex (code-davinci-002) on 50 samples                            | 60.9      | 23.1      | 50.2 | 15.3 |
> | GPT3 (text-davinci-002) on 50 samples                             | 62.6      | 28.8      | 61.3 | 23.1 |
> | ChatGPT (gpt-3.5-turbo) with 2 in-context examples on 100 samples | 59.9      | 46.0        | 17.8 | 10.0 |
> | ChatGPT (gpt-3.5-turbo) with 0 in-context example on 100 samples | 63.6 | 43.0 | 39.4 | 3.0 |

---

### Official Review · Reviewer_NBxP · 2023-08-05

**Soundness:** 3

**Excitement:**

3: Ambivalent: It has merits (e.g., it reports state-of-the-art results, the idea is nice), but there are key weaknesses (e.g., it describes incremental work), and it can significantly benefit from another round of revision. However, I won't object to accepting it if my co-reviewers champion it.

**Paper Topic And Main Contributions:**

This presents a simple approach to improve the performance of large language models on interactive decision making. It is tested on a web navigation dataset and beats sota.


**Questions For The Authors:**

Have you tested on other LLMs?
Have you tested modify the prompts in baseline methods to add sth equivalent to the summarizer and test its performances?

**Reasons To Accept:**

The proposed method is simple yet effective. The main contribution is a summaizer which can help remove noise. This partially mitigate the error propagation in long chains when using LLMs.

**Reasons To Reject:**

Only one LLM is tested here. It is not sure if the proposed method can be generalized to other LLMs.

**Reproducibility:**

3: Could reproduce the results with some difficulty. The settings of parameters are underspecified or subjectively determined; the training/evaluation data are not widely available.

**Reviewer Confidence:**

3: Pretty sure, but there's a chance I missed something. Although I have a good feel for this area in general, I did not carefully check the paper's details, e.g., the math, experimental design, or novelty.

---

> ### Author Rebuttal · Authors · 2023-08-29
>
> Thank you for your thorough reviews and valuable feedback. We understand the concern of testing only two older LLMs in the paper (the result for GPT-3 text-davinci-002 is provided in Appendix B), and have run our prompting method ASH on a more recent and chat-based LLM: ChatGPT (gpt-3.5-turbo). We choose this LLM to balance the strength and cost of running the experiment. We also chose the 4K token limit variant to test generalization to smaller token limit LLMs. While this initial analysis was run on a smaller product space with 1000 products and using 100 test samples, we will adopt the same sophisticated settings as the main results in the paper, and add it to the revised version of the paper along with the respective error analysis.
>
> As shown in the table below, ASH surpasses ReAct’s performance on ChatGPT, mirroring the same trend as when we were using the relatively weaker Codex models or GPT3. While the major merit of ASH in this setting has been its ability to condense observation allowing longer trajectories on the smaller token limit models, its ability to help agent recover from errors and align to the optimal behavior still holds as we observed on few sample test trajectories. The pronounced performance improvement with ASH can be attributed to the nature of ChatGPT agent: the agent attempts to buy the exact product and does not frequently settle with suboptimal products, compared to the older LLMs. We will also release the code and prompts for the chatgpt experiments in the same github repository as our main code.
>
> The main advantage of the proposed method is that the SUMMARIZER is orthogonal to the development of better ACTOR prompts, which the existing works have majorly focused on. Infact, our main method uses ReAct based ACTOR prompting conjugated with the SUMMARIZER. We have also demonstrated using a Act based ACTOR prompting and added SUMMARIZER in the loop. In future, we expect that more sophisticated ACTOR and SUMMARIZER prompts can be individually developed and finally combined to achieve superlative performances.
>
> | Model                                                             | **ASH**  Avg Score | **ASH** Success %     | **ReAct**  Avg Score | **ReAct** Success % |
> | ----------------------------------------------------------------- | --------- | --------- | --------- | --------- |
> | Codex (code-davinci-002) on 50 samples                            | 60.9      | 23.1      | 50.2 | 15.3 |
> | GPT3 (text-davinci-002) on 50 samples                             | 62.6      | 28.8      | 61.3 | 23.1 |
> | ChatGPT (gpt-3.5-turbo) on 100 samples | 59.9      | 46.0        | 17.8 | 10.0 |

---

### Meta-Review · Area_Chair_kJpA · 2023-09-20

**Recommendation:** 3

**Metareview:**

This manuscript proposes a method called Hierarchical Hint Assistant for optimizing the performance of LLMs on web navigation tasks. The effect of the proposed method is verified by experiments on the Webshop dataset.

Initially, reviewers were concerned about the representativeness of the LLMs tested. The authors addressed this issue by providing results on ChatGPT. Ablation results were also added in the rebuttal. The paper can be further improved by adding details of how SUMMARIZER works, details about the baseline models, and details of the experimental setup.

---

### Decision · Program_Chairs · 2023-10-07

**Decision:**

Accept-Findings

**Comment:**

This manuscript proposes a method called Hierarchical Hint Assistant for optimizing the performance of LLMs on web navigation tasks. The effect of the proposed method is verified by experiments on the Webshop dataset.

Initially, reviewers were concerned about the representativeness of the LLMs tested. The authors addressed this issue by providing results on ChatGPT. Ablation results were also added in the rebuttal. The paper can be further improved by adding details of how SUMMARIZER works, details about the baseline models, and details of the experimental setup.